# A reconstruction of warm water inflow to Upernavik Isstrøm since AD 1925 and its relation to glacier retreat.

Flor Vermassen[1,2], Nanna Andreasen[1,3], David J. Wangner[1,2], Nicolas Thibault[3], Marit-Solveig Seidenkrantz[4], Rebecca Jackson[1], Sabine M. Schmidt[5], Kurt H. Kjær[2], Camilla S. Andresen[1].

[1]Department of Glaciology and Climate, Geological Survey of Denmark and Greenland (GEUS), Copenhagen, Denmark

[2]Centre for GeoGenetics, Natural History Museum, University of Copenhagen, Copenhagen, Denmark;

[3]Department of Geosciences and Natural Resource Management, University of Copenhagen, Copenhagen, Denmark;

[4]Centre for Past Climate Studies, Arctic Research Centre, and iClimate Aarhus University Interdisciplinary Centre for Climate Change, Department of Geoscience, Aarhus University, Aarhus, Denmark;

[5]CNRS, OASU, EPOC, UMR5805, Pessac Cedex, France

*Correspondence to*: Flor Vermassen (flv@geus.dk)

**Abstract.** The mass loss from the Greenland Ice Sheet has increased over the past two decades. Marine-terminating glaciers contribute significantly to this mass loss due to increased melting and ice discharge. Periods of rapid retreat of these tidewater glaciers have been linked to the concurrent inflow of warm, Atlantic-sourced waters. However, little is known about the variability of these Atlantic-derived waters within the fjords, due to a lack of multi-annual, in situ measurements. Thus, to better understand the potential role of ocean warming on glacier retreat, reconstructions that characterize the variability of Atlantic water inflow to the fjords are required. Here, we investigate foraminiferal assemblages in a sediment core from Upernavik Fjord, West Greenland, in which the major ice stream Upernavik Isstrøm terminates. We conclude that the foraminiferal assemblage is predominantly controlled by changes in bottom water composition and provide a reconstruction of Atlantic water inflow to Upernavik Fjord, spanning the period 1925-2012. This reconstruction reveals peak Atlantic water influx during the 1930s and again after 2000, a pattern that is comparable to the Atlantic Multidecadal Oscillation (AMO). The comparison of these results to historical observations of front positions of Upernavik Isstrøm reveals that inflow of warm, Atlantic-derived waters likely contributed to high retreat rates in the 1930s and after 2000. However, moderate retreat rates of Upernavik Isstrøm also prevailed in the 1960s/1970s, showing that glacier retreat continued despite a reduced Atlantic water inflow, albeit at a lower rate. Considering the link between bottom water variability and the AMO in Upernavik Fjord, and the fact that a persistent negative phase of the AMO is expected for the next decade, Atlantic water inflow into the fjord may decrease in the coming decade, potentially minimizing or stabilizing the retreat of Upernavik Isstrøm during this time interval.

# 1 Introduction

Mass loss from the Greenland Ice Sheet (GrIS) has accelerated over the two recent decades, raising its contribution to the ongoing global sea-level rise. Currently, about one third of the mass loss is attributed to dynamic loss in the form of ice discharge from large tidewater glaciers (van den Broeke et al., 2016). The processes controlling the instability of these glaciers are however still not well understood, and therefore the capability of computational models to predict the rate of future mass loss remains limited. During the early 2000s a rapid retreat of tidewater glaciers along the south-eastern (SE) sector (Rignot et al., 2004), Central West sector (Holland et al., 2008) and northwest (NW) sector (Bjørk et al., 2012; Joughin et al., 2013; Khan et al., 2010) was observed. This retreat coincided with a warming of the ocean waters in SE and W Greenland and led to the hypothesis that shifting ocean currents exert a major control on the dynamic behaviour of these glaciers (Holland et al., 2008; Straneo et al., 2013). Several studies have examined past ice-ocean interactions to investigate the importance of ocean forcing on a longer timescale (Andresen et al., 2010, 2011, 2017; Dyke et al., 2017; Lloyd et al., 2011; Wangner et al., 2018) but few have the temporal resolution needed for an in-depth study of (sub-)decadal dynamics (Andresen et al., 2012b; Drinkwater et al., 2014). In West Greenland specifically, only a few observations or detailed reconstructions of bottom water temperatures throughout the 20th century exist (Lloyd et al., 2011; Ribergaard et al., 2008). Sediment records in Greenlandic fjords with marine-terminating glaciers are usually characterised by relatively high sedimentation rates (Dowdeswell et al., 1998), thus allowing environmental reconstructions with a high temporal resolution. Benthic foraminiferal communities are particularly sensitive to environmental conditions, and shifts in species composition can be used to reconstruct relative changes of bottom water masses (Murray, 2006).

Here, we investigate foraminiferal assemblages in a marine sediment core from Upernavik Fjord, NW Greenland. We evaluate which environmental factors influence the foraminiferal assemblages and reconstruct Atlantic water inflow to Upernavik Isstrøm since 1925. These results are compared to historical records of the Upernavik Isstrøm ice front retreat in order to evaluate whether periods of increased Atlantic water inflow in the fjord are associated with glacier retreat.

# 2 Study area and previous research

Upernavik Fjord has a length of ~60 km and is 5-7 km wide. The fjord floor has been mapped with a Multibeam Echo Sounder System (MBES) as part of NASA's Oceans Melting Greenland mission (NASA OMG Mission, 2016). The outer part of the fjord is characterised by steep side walls and a flat fjord floor at ~900m water depth (Fig. 1).

Recent CTD measurements have revealed a stratified water column in Upernavik Isfjord with evidence for warm-water entrainment at depth near the glacier front (below ~250m) (Andresen et al., 2014; Fenty et al., 2016). This warm-water layer flows below a colder and fresher surface water layer. The stratification is the result of two ocean currents that influence the hydrography of the fjord (Fig. 1). The West Greenland Current (WGC) flows northward as a subsurface current along the West Greenland margin and transports relatively warm, saline waters (mean temperature >4°C, mean salinity >34.91 psu; Myers et al., 2007). The West Greenland Current originates from the Irminger Current (IC) in the Atlantic Ocean (Fig. 1). Along its

path from east to west, the Irminger current (and subsequently West Greenland Current) mix with colder and fresher waters of the East Greenland Current, which transports cold and less saline waters from the Arctic Ocean as a near-surface current (<1°C, <34.9 psu; Sutherland and Pickart, 2008). Local waters, heavily influenced by meltwater from the Greenland Ice Sheet, are found at the very surface of the West Greenland Current. The West Greenland Current reaches Upernavik Fjord via a deep

trough (~700m) (Fig. 1).

Upernavik Isstrøm is a major ice stream in NW Greenland, draining a significant part of the Greenland Ice sheet (~65,000 km$^2$) (Haubner et al., 2018) (Fig. 1). In 1886, Upernavik Isstrøm was characterised by a single glacier front, but since then it has retreated into different branches of the fjord (Weidick et al. 1958). Currently, four glaciers calve into the fjord waters (named Upernavik 1 to 4, Fig. 2).

Upernavik Isstrøm is one of few outlet glaciers in Greenland with historical observations of its ice-front position throughout the 20[th] century. A compilation of these shows retreat of Upernavik Isstrøm starting in 1930 (Khan et al., 2013; Weidick, 1958). These observations were used to prescribe a 3D model that estimated mass loss between 1849-2012 (Haubner et al., 2018). This revealed a period with near zero mass loss between 1849–1932, mass loss dominated by ice dynamical flow between 1932-1998, and finally a total mass loss due to an increased negative surface mass balance as well as increased ice

dynamical loss, with a magnitude twice that of any earlier period between 1998-2012. A study of aerial photographs revealed a shift in mass loss dominated by thinning of the southern glaciers between 1985-2005, to mass loss mainly due to thinning of the northern glaciers between 2005-2010 (Kjær et al., 2012). Asynchronous behaviour of the different glaciers after 2005 was also described by Larsen et al. (2016); this study showed an acceleration of Upernavik 1 and Upernavik 2 in 2006 and 2009, respectively, while the southern glaciers Upernavik 3 and 4 remained stable. Sediment cores from Upernavik fjord were

investigated for their IRD content and compared to historical glacier front observations (Vermassen et al., 2019). This study demonstrated that the spatial variability of IRD patterns are high, and that randomness inherent to ice-rafting largely overprints the glaciological signal (i.e. iceberg calving) within this fjord. Their use as an indicator of glacier stability was improved by producing a composite record based on multiple cores, but due to the complexities associated with the interpretation of the IRD record we focus on comparing the results of the benthic foraminifera analysis with historical observations of glacier

margins in this study.

## 3 Methods

### 3.1 Core collection

Sediment core POR13-05 (172 cm, 72.945 N, 55.620 W) was collected from 900 m water depth in August 2013 during a cruise

with *R/V Porsild*. Coring was undertaken with a Rumohr corer (Meischner and Ruhmohr, 1974). This type of device is specifically designed to avoid sediment disturbance during coring, thereby ensuring preservation of the core top. This study focuses on the top 50 cm of the core.

## 3.2 Grain-size analysis

The core was sampled continuously every centimetre for grain size analysis. In order to calculate the water content, sample weight was measured before and after freeze-drying. Wet-sieving was performed on all samples, separating them into three grain-size fractions (<63 µm, 63-125 µm, and >125 µm). For the >125 µm fraction, singular pebbles >0.01g were discarded from the measurements to avoid distortion of the analysis due to the occurrence of an individual large grain (Wangner et al., 2018). The different fractions were weighed and the individual percentages were calculated relative to the total dry weight. A detailed discussion and methodology of IRD variability can be found in Vermassen et al. (2019).

## 3.3 Age model

Sediment ages were determined with $^{210}$Pb dating. Seven samples each obtained from a 1 cm interval were freeze-dried and 4–6 g of dried sediment was then conditioned in sealed vials. A well-type gamma detector Cryocycle-I (Canberra) at the laboratory UMR5805 EPOC (University of Bordeaux, France) was used to measure $^{210}$Pb, $^{226}$Ra and $^{137}$Cs. Estimated errors of radionuclide activities are based on 1 standard deviation counting statistics. Excess $^{210}$Pb ($^{210}$Pb$_{xs}$) is calculated by subtracting the activity supported by its parent isotope, $^{226}$Ra, from the total $^{210}$Pb activity in the sediment. The CF-CS (constant flux, constant sedimentation) model was applied to calculate a sedimentation rate. The sedimentation rate was then used to calculate sediment ages. The sediment surface was assumed to represent the year of core acquisition (2013).

## 3.4 Foraminifera analysis

Sediment slices (1 centimeter) were sub-sampled for foraminiferal studies (6-13 g of wet sediment), so that they each contained an estimated amount of ~300-~500 tests. Foraminiferal assemblage analysis was performed on 26 samples from the top 50 cm of the core. This corresponds to a time-resolution of ~4 years between samples. The fresh samples were soaked overnight in a light alkaline solution (Na$_2$CO$_3$, 15g/l) to disintegrate silt and clay clumps in the samples. Subsequently, they were wet-sieved with a 63 µm sieve and stored with a storage solution, consisting of distilled water, ethanol and sodium carbonate. In order to preserve the most fragile species, the foraminifera were wet-counted (Bergsten, 1994) together with foraminiferal organic linings. Partially dissolved tests were counted as organic linings if the tests did not display enough features to allow species identification. A minimum of 300 tests were counted in each sample. Based on previous research, species were categorised into three groups: (chilled) Atlantic water indicators, Arctic water indicators, and those with no apparent specific environmental preference ('indifferent'). We use these previously proposed categorisations to explore whether these groupings are also evident in our dataset (i.e. whether species with similar environmental preferences display similar variability in Upernavik Fjord). It should be noted that for some species, different environmental preferences have been suggested in the literature (Table 1). In those cases, we categorise the species according to the most recent literature and in accordance with sites most comparable to our study, but also list the contradicting references (Table 1). A principal component analysis (PCA) of species abundances >0.5% was performed with the software PAST (Hammer et al., 2001). This was done to simplify analysis

and visualise variation within the dataset. Abundances were normalised before PCA analysis by calculating z-scores to avoid skewing of the data by species with large abundances. Correlations of time series were calculated in Microsoft Excel using the Pearson correlation function.

## 4 Results

### 4.1. Core lithology and sedimentology

Visual observations of the core section reveal a brown mud (code 10 R 5/6 (Munsell, 1912)) with sub-angular to sub-rounded pebbles interspersed throughout the core (Fig. 3). The split core surface shows no sign of bioturbation, as also confirmed by the X-ray image and the age model. Grain-size measurements and the X-ray image show that the lower part (172-90 cm) of core POR13-05 is characterised by an alternation of mud-supported diamicton versus homogeneous mud without pebbles. The mid-section of the core contains a thick, sandy turbidite, characterized by a fining-upwards trend (90–70 cm), capped by a muddy, pebble-free unit at 68-53 cm (Fig. 3). The upper-part of the core (53–0 cm) is composed of massive, mud-supported diamicton, predominantly comprising of clay (>90 %) interspersed with larger clasts up to pebble size.

### 4.2 Age model

The 50 cm display a logarithmic decay of $^{210}$Pb, indicating continuous sedimentation (Fig. 4). Based on the CF-CS model a sedimentation rate of 0.58 cm/yr was derived. This corresponds to an age of 1925 ± yr at 50 cm depth. The turbidite at 92-65 cm hinders dating further downcore.

### 4.3 Foraminifera assemblage variability

A total of 35 benthic foraminiferal species were identified, of which 16 species are calcareous and 19 agglutinated (see Table 1 for full species names). The planktonic foraminifera *Neogloboquadrina pachyderma* (sinistral) is only present in two distinct intervals, between 1-8 cm and 44-50 cm, albeit at very low abundances (<5 counts/g).

Due to relatively low counts, the percentage-distribution of each species is calculated relative to the total benthic assemblage, i.e. both calcareous and agglutinated specimens (Fig. 5). The species representing >0.5% of the total assemblage are presented. However, the counts and percentages of species with lower abundances can be found in the supplementary material (Table S2). As the calcareous vs. agglutinated foraminiferal concentrations may also be affected by dissolution of calcareous tests and post-mortem degradation of agglutinated specimens, we also show percentage data of calcareous species relative to only the calcareous fauna and agglutinated species only relative to the agglutinated assemblages in the supplementary material (Fig. S1). Samples with <70 specimens are excluded from these calculations due to the high uncertainties related to low count numbers (Fig. S1).

We base our habitat categorisation on previous studies from Disko Bugt (Lloyd et al., 2007, 2011, Perner et al., 2011, 2013; Wangner et al., 2018), the central West-Greenland slope (Jennings et al., 2017) and the continental shelf (Sheldon et al., 2016). The agglutinated assemblage >0.5% comprises 1 species that is indicative of Atlantic waters (*A. glomerata*), the others are indicative of Arctic waters or are non-indicative (Fig. 5, Table 1). Within the calcareous assemblage (>0.5%), 2 species are

indicative of (chilled) Atlantic waters (*N. labradorica*, *N. auricula),* 3 of Arctic water conditions (*S. feylingi, S. concava, E. clavatum*), and 2 are non-indicative (*C. reniforme, B. pseudopunctata*) (Fig. 5, Table 1).

Agglutinated species generally dominate the assemblage and range between 45-99% of the total assemblage. Among these, *T. earlandi* and *S. biformis* are the most abundant species with a median of 17% and 14%, respectively. The other agglutinated species are markedly less abundant with medians < 5%. The most abundant calcareous species are *S. feylingi* and *E. clavatum*,

with a median of 16% and 4% of the total count, respectively. The median percentage of other calcareous species is lower than 2%.

The variability in species assemblage is broadly characterised by three intervals. Between 50 and 25 cm, the percentage of calcareous species is high, varying between 40-70% (Fig. 5). Between 25 and 12 cm, calcareous abundances are low (<20%) with almost none present between 16-12 cm. Between 12 and 0 cm, the amount of calcareous species increases again up to

40%, and high values of 40-50% occur between 6 and 0 cm. The concentration of foraminifera, expressed as counts/gram (g) of wet sediment, varies generally between 60-150 counts/g. Outliers with anomalous low and high counts/g are present at 45 cm (21 counts/g) and 1 cm (308 counts/g), respectively. The amount of organic linings per gram shows a trend that is inversely proportional to the percentage of calcareous species. Since very few calcareous species are present between 7-22 cm, this interval does not allow potential changes in bottom waters masses to be estimated from the calcareous assemblage. Within the

agglutinated fauna (>0.5%), there is only one Atlantic Water indicator, which also questions whether shifts *within* the agglutinated assemblage alone can be used as a reliable reconstruction of Atlantic vs. Arctic Water masses. Also, ecological preferences of agglutinated species have received less regard than those of calcareous fauna.

### 4.4 PCA analysis

Principal components 1 and 2 explain 28.6% and 16.8% of the variance in the dataset, respectively. The loadings plot reveals a clustering of three main groups of foraminiferal species (Fig. 6). The first group is characterised by strongly negative PC1 loadings (group 1). This group consists of calcareous species, both (chilled) Atlantic Water and Arctic indicators. The second group consists of cold-water agglutinated species, characterised by strongly positive PC1 loadings (group 2). The third group consists mostly of species that are not indicative of a specific water mass (group 3). These species have low loadings for PC1

and strongly negative loadings for PC2. These results show that variability within the dataset can be predominantly explained by shifts between calcareous vs. agglutinated species (i.e. PC1). However, to investigate relative shifts *within* the respective assemblages, we also show abundances and PCA analysis relative to wall composition (Fig S2 and S3).

# 5 Discussion

## 5.1 Environmental controls on species abundance in Upernavik Fjord

Foraminifera are sensitive to changes in water depth, substrate, bottom water characteristics (temperature, salinity) and nutrient flux (Lloyd, 2006b; Murray, 2006). Species displaying the same pattern of variability respond to similar environmental changes. To determine which environmental parameters control species abundances in Upernavik Fjord, we compare our PCA results with the ecological preferences of species as suggested in the literature.

PC 1 is dominated by the opposing trends observed in the abundance of agglutinated and calcareous species (Fig. 6). The fact that the agglutinated species *T. torquata, S. biformis, P. bipolaris,* and *R. turbinatus* show similar trends is expected since the literature suggests they thrive in similar environmental conditions, i.e. cold, Arctic waters (Table 1). It should be noted that even though previous studies attributed these species to Arctic waters, they have also been identified in sediments in Gullmar Fjord, SE Sweden (Höglund, 1947; Polovodova Asteman et al., 2013). Therefore, we add a cautionary note that these species could in fact be more cosmopolitan than commonly assumed. Nevertheless, to remain consistent with recent studies in West Greenland we consider them here indicative of Arctic waters.

In the calcareous assemblage the Arctic-water indicators *S. concava* and *S. feylingi* display similar trends as the (chilled) Atlantic water indicators *N. labradorica* and *N. auricula* (Lloyd, 2006b; Lloyd et al., 2007; Wangner et al., 2018) . Similar trends in these species could also be because they have all (except *N. auricula*) been suggested as indicators of high-productivity environments, i.e. they thrive when the flux of organic material from surface waters is high (Jennings et al., 2017 and references therein). Similarity with the pattern of abundance of *B. pseudopunctata*, a species that is considered indifferent to changes in Atlantic/Arctic waters but sensitive to surface water productivity (Sheldon et al., 2016), further supports this interpretation. Thus, the PCA results suggest that the assemblage diversity is influenced both by environmental factors that control the ratio of agglutinated/calcareous species, as well as by changes of the primary productivity in the fjord.

The ratio between calcareous and agglutinated foraminifera is often used as a proxy for environmental change, as these two groups have different environmental preferences and preservation potential; large shifts between agglutinated and calcareous species are commonly seen in Arctic records (Andresen et al., 2012a; Seidenkrantz et al., 2007). An important factor controlling the abundance of calcareous foraminifera is the concentration of $HCO_3^{2-}$ in the ambient water. In cold waters the calcification process may be limited by a lack of $HCO_3^{2-}$, while warmer waters promote calcification. In addition, and perhaps more importantly, the preservation of calcareous specimens can be influenced by post-mortem dissolution. Colder waters contain more $CO_2$ and are thus less alkaline than warmer waters; this promotes dissolution of calcareous tests. Test dissolution is common in Arctic waters and appears similarly important in Upernavik Fjord, implied by the strong variation of the total calcareous abundance and an interval dominated by agglutinated fauna (7-22 cm). The fact that post-mortem dissolution plays an important role in explaining the patterns of our dataset is further supported by the inverse relationship between the percentage of calcareous species and foraminiferal test linings (Fig. 6). Organic linings are less susceptible to dissolution than calcareous tests and remain better preserved under corrosive conditions. Hence, during periods with relatively warm waters

entering the fjord, carbonate tests are better preserved, whereas colder water conditions will result in a higher abundance of organic linings. Warm subsurface (bottom) waters prevail in the fjord when the Atlantic-derived portion of the West Greenland Current is high. Conversely, when the Atlantic portion of the West Greenland Current is low and contribution of Arctic-sourced waters (i.e. East Greenland Current) is high, colder bottom waters will corrode calcareous benthic foraminiferal tests.

Therefore, we suggest that the abundance of calcareous foraminifera represents here a proxy for the inflow of warm, Atlantic-derived waters to the fjord. This interpretation is strengthened by a similar study in the Ameralik Fjord (SW Greenland), where the percentage of calcareous foraminifera was used as a proxy for the influx of warm, Atlantic-derived waters (Seidenkrantz et al., 2007; Seidenkrantz, 2013). In that study, it was suggested that calcification of benthic foraminiferal tests is prevented at times of sea-ice growth due to the associated formation of $CO_2$ rich brine waters. In Upernavik Fjord we also suggest an

influence of bottom water alkalinity on foraminifer assemblage, but it should be noted that the mechanism by which corrosive cold water currents circulate in the fjord is probably different. In contrast to Ameralik Fjord, the absence of a sill separating Upernavik Fjord from the Baffin Bay allows a more direct connection to the open ocean. This permits shelf currents to enter the fjord, and together with outflowing surface meltwater this results in a strong circulation in the fjord (buoyancy-driven circulation; Cowton et al., 2016). Thus, bottom waters are likely well-ventilated and influenced directly by inflowing bottom

waters rather than by the sinking of brine waters. It is important to note, however, that high sedimentation rates may shield calcareous fauna from dissolution; fast burial of benthic fauna limits the time available for dissolution by bottom waters and this has been observed in the glaciomarine environment (Lloyd, 2006a; Lloyd et al., 2005). Since the age model of core POR13-05 reveals a constant sedimentation rate, variations in sedimentation rate probably do not affect preservation of the calcareous fauna in this study.

Furthermore, we propose a possible influence of Atlantic waters on the nutrient level in the fjord, which in turn affects the benthic calcareous species assemblage. Recent research showed that primary productivity in a fjord with a marine-terminating glacier is predominantly controlled by rising subsurface meltwater plumes entraining ambient nutrient-rich deep water to the surface (Meire et al., 2017). Productivity in a fjord with a land-terminating glacier (Young Sound, NE Greenland) and one with marine-terminating glaciers (Godthåbsfjord, SW Greenland) were compared, revealing that the limiting amount of

nutrients available for phytoplankton in these fjords is predominantly delivered through upwelling of Atlantic-derived waters (Meire et al., 2017). Periods with more Atlantic water inflow would thus lead to a higher primary productivity, which in turn would favour meso- and eutrophic benthic species at the sea-floor. Such a mechanism might explain the similar abundance trends of the Atlantic indicators *N. labradorica* and *N. auricula* together with the cold-water productivity indicators *S. feylingi* and *S. concava* and with the mesotrophic indicator *B. pseudopuncata*, all species essentially responding here to variations in

bottom-water nutriency (Fig. 5 and 6). However, these inferences with regard to nutrient level in the fjord and its relation to benthic fauna should be substantiated with additional proxies that reconstruct primary productivity, and future research in glacial fjords is required to confirm this interpretation.

The planktonic foraminifera *N. pachyderma* (sinistral) occurs concurrently with high abundances of benthic calcareous species in POR13-05 and also simultaneously with the highest temperatures observed in the West Greenland current (Ribergaard et

al., 2008), i.e. during AD 1925-1935 AD and after AD 2000 (Fig. 8). *N pachyderma* is a polar species that typically dominates the planktonic assemblage high latitude environments (Schiebel et al., 2017). However, planktonic foraminifera in general do not tolerate reduced salinities and the Arctic types of *N. pachyderma* seems to avoid salinities <32 psu (Carstens et al., 1997). The presence of planktonic foraminifera may thus be considered a proxy for the influx of oceanic water. In Upernavik Fjord,

the upper water layer is enriched by glacial meltwater, leading to a relatively fresh upper water layer (Fenty et al., 2016) and to generally unfavourable conditions for planktonic foraminifera. Hence, we suggest that when the Atlantic-derived water layer increases and thickens, reaching shallower depths and increasing salinity levels, this may allow *N. pachyderma* (sinistral) to inhabit Upernavik Fjord. A similar scenario has been described in northern Baffin Bay, where changes in the abundance of *N. pachyderma* (sinistral) has been linked to variable freshwater flux and associated salinity changes (Knudsen et al., 2008).

In summary, we suggest that the dominant control on the composition of the foraminiferal assemblage essentially relates to (post-mortem) dissolution of calcareous species, controlled by variations in the alkalinity of the West Greenland Current and associated contribution of Atlantic water. As a potential second control, an increased inflow of Atlantic waters is associated with upwelling of nutrients and thus higher primary productivity, which could favour benthic species that thrive under a high supply of organic matter from surface waters. The latter control should be confirmed in future research. Based on these results,

we use total calcareous foraminiferal concentration as a proxy for Atlantic water inflow to Upernavik Fjord. The presence of *N. pachyderma* (sinistral) associated with the maximal West Greenland Current temperatures may also be used to reconstruct the inflow of Atlantic-sourced water.

**5.2 Comparison of Atlantic water inflow with climatic records**

The foraminiferal record indicates cold bottom water masses in the early 1920s, followed by a rapid warming that peaks in

the mid 1930s and subsequently cooling during the 1940s (Fig. 7). After the 1940s relatively warm bottom water masses prevail, followed by a strong cooling that commences in the mid 1960s and reaches minimal values between the mid 1970s and 1990. In the early 1990s a rapid warming occurs, reaching peak values after 2005. Our reconstructed record is supported by the measured bottom water temperatures at Fylla Banke (400-600m), located offshore Nuuk in SW Greenland (Fig. 1 and 7; Ribergaard et al., 2008), which shows a similar pattern. This gives confidence in our reconstruction and confirms that bottom

water changes in Upernavik Fjord are linked to the variability of the West Greenland Current. The presence of *N. pachyderma* (sinistral) during 1920s-mid 1930s and after 2000 could suggest that the inflow of Atlantic water into Upernavik Fjord was more intense during these periods than during 1950-1965, when relatively high amount of calcareous species are present but *N. pachyderma* (sinistral) is absent.

A previous foraminifera-based reconstruction of Atlantic-water influx in Disko Bugt (Lloyd et al., 2011) shows an overall

comparable Atlantic Water inflow pattern, except for the period 1950-1975, during which our reconstruction of warm bottom water masses in Upernavik contrasts with the colder conditions in Disko Bugt (Fig. 7). Potentially, this is related due to age uncertainties in both records, or subtle differences in the sensitivity of the foraminiferal assemblages to the prevailing bottom water mass.

The pattern of bottom water variations reconstructed here is also similar to the pattern of the Atlantic Multidecadal Oscillation (NOAA ESRL, 2018; Enfield et al., 2001), with a Pearson's correlation coefficient r=0.42 and significant at p=0.05. The AMO is a 55-70 year cyclical pattern in Atlantic water temperature, usually considered as an expression of the variability of the overall Atlantic Ocean circulation (Kerr, 2000; Knudsen et al., 2011; Schlesinger and Ramankutty, 1994; Trenberth and Shea, 2006). A link between the physical oceanography of West Greenland and Atlantic SSTs has indeed been suggested previously: a positive phase of the AMO is related to an increase of warm Atlantic waters flowing towards and along the SE and W Greenland shelf (Drinkwater et al., 2014; Lloyd et al., 2011). Our data supports that the AMO is related to bottom water temperature variability along the West Greenland shelf and shows that this influence is strong within Upernavik Fjord.

## 5.3 Retreat of Upernavik Isstrøm and ocean forcing

The retreat history of Upernavik Isstrøm is relatively well constrained from historical observations (Fig. 2), although some care should be taken in interpreting this record because of the relatively low temporal resolution for the older parts of the record. By comparing this record to our reconstruction of bottom water variability, we evaluate whether ocean warming was concurrent with periods of retreat throughout the 20[th]/21[st] century (Fig 7).

Despite differences in the timing and magnitude of the retreat of the different glaciers, they broadly share the same retreat history. High retreat rates occurred between the mid '30s and mid '40s (400-800m/yr), moderate retreat rates between 1965-1985 (~200 m/yr, except for Upernavik 3) and high retreat rates again after 2000 (>200 m/yr) (Fig. 8). The largest difference between individual glaciers occurred after 2005, when the northern glaciers Upernavik 1 and Upernavik 2 retreated and thinned fast, but the southern glaciers Upernavik 3 and Upernavik 4 remained relatively stable (Kjær et al., 2012; Larsen et al., 2016) (Fig. 8).

Our reconstruction reveals that relatively warm, Atlantic-derived bottom waters prevailed in Upernavik Fjord in the 1930s and after 1995. Warm water inflow may have triggered the substantial retreat that was observed during these periods. However, the cause of these periods of retreat cannot be attributed solely to ocean forcing since air temperatures mostly co-vary with the reconstructed bottom water variability (Fig. 8). Disentangling the relative importance between oceanic and atmospheric forcing remains challenging (Straneo and Heimbach, 2013). Both processes are inherently linked not only due to a common regional forcing of both (AMO), but also because increased glacier run-off (due to warmer air temperatures) can strengthen fjord circulation and thus increase inflow of Atlantic waters (Carroll et al., 2016; Christoffersen et al., 2011). Moderate retreat rates during 1960-1985 coincided with cooling of bottom waters, showing that retreat occurred despite a stabilising effect from decreasing ocean water temperatures. Modelling of the ice thickness and velocity changes of Uperavik Isstrøm showed that retreat during this period was most likely perpetuated by dynamic ice discharge (Haubner et al., 2018), which is in part the result of changes in glacier dynamics due to variations in bed topography. Bed topography can play indeed play a major role on glacier retreat; on reverse bed slopes glaciers are considered unstable because of the increased ice discharge associated with retreat into deeper bedrock (Jamieson et al., 2012; Nick et al., 2013). A conclusive evaluation of the role of bedrock topography

during this timeframe would require a higher temporal resolution of glacier front observations, however, and is not within the scope of this study.

Altogether, our findings emphasise that the timing and magnitude of the retreat of Upernavik Isstrøm throughout the 20th/21st century is not simply a function of bottom water temperature variability but reflects a complex response to multiple mechanisms.

Finally, we note that even when the inflow of Atlantic water to the fjord is high, variations in fjord bathymetry can determine whether the warm waters are able to reach the glacier front(s). This could, for example, explain the differential response of the northern glaciers (Upernavik 1 and 2) versus the southern glaciers (Upernavik 3 and 4) after 2000 AD (Andresen et al., 2014). The northern glaciers (Upernavik 1 and 2) are positioned on a deep bed (~900m), whereas the southern glaciers (Upernavik 3 and 4) are positioned on a shallower bed (<400 m water depth). Because the Atlantic water layer flows at 400-900 m depth, the southern glaciers would not have been affected by warming of the Atlantic layer, in contrast to the northern glaciers.

## 6 Conclusions

In this study, we show that the abundance and diversity of foraminiferal species in the outer region of Upernavik Fjord is predominantly controlled by the preservation potential of calcareous species, depending on the alkalinity of the prevailing bottom water mass, which is in turn related to variations in water temperature. Therefore, we use the percentage of calcareous species as a proxy for the inflow of the warm, Atlantic component of the West Greenland Current. This reconstruction spans the period 1925-2012 and broadly displays the same pattern of variability as the Atlantic Multidecadal Oscillation. Comparison of our bottom water record with historical observations of glacier front positions reveals that warm subsurface waters were associated with periods of increased retreat rates of Upernavik Isstrøm during the 1930s and after 1995 AD. Conversely, moderate retreat rates of Upernavik Isstrøm during 1960-1985 was associated with cooling bottom waters, showing that this retreat was not simply a function of bottom water variability. Thus, our study shows that while warming of ocean waters in Upernavik fjord likely contributed to the retreat phases during the 1930s and early 2000s, ocean warming is not a prerequisite for retreat of Upernavik Isstrøm. The similar pattern of our bottom water reconstruction and the Atlantic Multidecadal Oscillation shows that ocean currents interacting with Upernavik Isstrøm depend on ocean circulation changes originating in the North Atlantic, a finding that is consistent with a study in Disko Bugt, located ~450km further south of our study site (Lloyd et al., 2011). This is important since it implies that the future potential oceanic forcing of Upernavik Isstrøm can be assessed from observed circulation changes in the North Atlantic. Since the Atlantic meridional overturning circulation strength and associated heat transport is currently declining (Frajka-Williams et al., 2017), this may lead to cooling of the West Greenland during the next decade. As a result, this cooling could potentially and temporarily minimise the retreat of Upernavik Isstrøm and other marine-terminating glaciers along the West Greenland coast, as has been recently observed for Jakobshavn Isbræ (Khazendar et al., 2019).

## 7 Acknowledgements:

This study is a contribution to the VILLUM FONDEN project 'Past and Future Dynamics of the Greenland Ice Sheet: what is the ocean hiding?' (10100).

MSS was supported by the Independent Research Fund Denmark (G-Ice project Grant No. 7014-00113B/FNU).

## 8 Data availability

Foraminiferal abundances and age model are available at the Pangea data archive. https://doi.org/10.1594/PANGAEA.896945.

## 9 Conflicts of interest

Marit-Solveig Seidenkrantz is a co-editor-in-chief for Climate of the Past but was not involved in the editorial process of this article.

## 10 Author contributions

FV interpreted all data, performed sediment lab work, and wrote the manuscript. NA counted foraminifer species and
contributed to data interpretation. RJ and DJW helped with identification of the species and contributed to analysis of the data. NT contributed to data interpretation. CSA and KK designed the study. SS developed the age model. MSS revised the identification of the foraminiferal species and contributed to their interpretation. All co-authors contributed to the writing of the manuscript.

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

## 12 Tables

**Table 1**: overview of environmental preferences of the benthic foraminiferal species identified in this study.

| Agglutinated species | First description | Atlantic influenced | References | Arctic | References | Indifferent | References | Productivity | References |
|---|---|---|---|---|---|---|---|---|---|
| *Adercotryma glomerata* | (Brady, 1878) | x | Lloyd (2006), Lloyd et al. (2011), Perner et al., (2013), Sheldon et al. (2016), Wangner et al. (2018) | | | | | | |
| *Cribrostomoides crassimargo* | (Norman, 1858) | | | x | Sheldon et al. (2016) | x | Lloyd (2006), Lloyd et al. (2011), Perner et al. (2011, 2013), Wangner et al. (2018) | | |
| *Cuneata arctica* | (Brady, 1881) | | | x | Lloyd, (2006), Lloyd et al. (2011), Sheldon et al. (2016) | | | | |
| *Portatrochammina bipolaris* (often described as *Portatrochammina karica*) | Brönnimann & Whittaker, 1980 | | | x | Jennings et al. (2017) | | | | |
| *Reophax catella* | Höglund, 1947 | x | Jennings et al. (2017), Sheldon et al. (2016) | | | x | Wangner et al. (2018) | | |
| *Reophax catenata* | Höglund, 1947 | x | Jennings et al. (2017) | | | x | Wangner et al. (2018) | | |
| *Reophax subfusiformis* | Earland, A. 1933 | x | Jennings et al. (2017), Wangner et al. (2018) | | | | | | |
| *Lagenammina difflugiformis / Reophax difflugiformis*) | (Brady, 1879) | x | Perner et al. (2013), Jennings et al. (2017), Sheldon et al. (2016), Wangner et al. (2018) | | | | | | |

| Agglutinated species | First description | Atlantic influenced | References | Arctic | References | Indifferent | References | Productivity | References |
|---|---|---|---|---|---|---|---|---|---|
| *Spiroplectammina biformis* | (Parker & Jones, 1865) | | | x | Lloyd (2006), Lloyd et al. (2011), Perner et al. (2011, 2013), Jennings et al. (2017), Wangner et al. (2018) | | | | |
| *Textularia earlandi* | Parker, 1952 | x | Jennings et al. (2017) | x | Lloyd (2006), Lloyd et al. (2011), Jennings et al. (2017), Sheldon et al. (2016), Wangner et al. (2018) | | | | |
| *Cuneata arctica* | (Brady, 1881) | | | x | Lloyd (2006), Lloyd et al. (2011), Perner et al. (2011), Perner et al. (2013), Jennings et al. (2017), Wangner et al. (2018) | | | | |
| *Textularia torquata* | Parker, 1952 | | | x | Perner et al. (2013), Sheldon et al. (2016), Wangner et al. (2018) | x | Lloyd (2006), Lloyd et al. (2011), Perner et al. (2011) | | |

| Calcareous species | First description | Atlantic influenced | References | Arctic | References | Indifferent | References | Productivity | References |
|---|---|---|---|---|---|---|---|---|---|
| *Bolivina pseudopuncata* | (Höglund, 1947) | x | Lloyd (2006) | | | x | Lloyd et al. (2011), Perner et al. (2011), Perner et al. (2013) | x | Sheldon et al. (2016) |

| Species | Author | | | | | | | | |
|---|---|---|---|---|---|---|---|---|---|
| *Buccella frigida* | (Cushman, 1922) | x | Jennings et al. (2017) | | | x | Lloyd et al. (2011), Perner et al. (2011), Perner et al. (2013) | x | Jennings et al. (2017) |
| *Cassidulina reniforme* | Cushman, 1930 | x | Lloyd et al. (2011), Perner et al. (2011), Perner et al. (2013), Jennings et al. (2017) | x | Sheldon et al. (2016) | | | | |
| *Cibicides lobatulus* | (Walker & Jacob, 1798) | | | | | x | Lloyd (2006), Lloyd et al. (2011), Perner et al. (2011), Perner et al. (2013) | | |
| *Elphidium clavatum* | Cushman, 1930 | | | x | Lloyd et al. (2011), Perner et al. (2011), Perner et al. (2013), Jennings et al. (2017), Sheldon et al. (2016) | x | Wangner et al. (2018) | | |
| *Epistominella takayanagi* | Iwasa, 1955 | | | | | | | | |
| *Globobulimina auriculata arctica* | Höglund, 1947 | | | | | x | Perner et al. (2013) | | |
| *Islandiella helenae* | Feyling-Hanssen & Buzas, 1976 | | | x | Lloyd (2006), Perner et al. (2011), Perner et al. (2013), Jennings et al. (2017), Wangner et al. (2018) | | | x | Jennings et al. (2017) |
| *Islandiella norcrossi* | (Cushman, 1933) | x | Perner et al. (2011), Perner et al. (2013), Jennings et al. (2017), Sheldon et al. (2016), Wangner et al. (2018) | x | Lloyd (2006) | | | | |
| *Melonis barleeanus* | (Williamson, 1858) | x | Lloyd (2006), Jennings et al. (2017), Sheldon et al. (2016), Wangner et al. (2018) | | | x | Perner et al. (2013) | x | Jennings et al. (2017), Sheldon et al. (2016) |
| *Nonionellina auricula* | Heron-Allen & Earland, 1930 | | Lloyd et al., (2006) Wangner et al., (2018) | | | | | | |

| Species | Author | | | | | | | | |
|---|---|---|---|---|---|---|---|---|---|
| *Nonionellina labradorica* | (Dawson, 1860) | x | Lloyd (2006), Wangner et al. (2018) | | | x | Lloyd et al. (2011), Perner et al. (2011), Perner et al. (2013) | x | Jennings et al. (2017), Sheldon et al. (2016) |
| *Nonionellina turgida* | Williamson, W.C. (1858) | x | Jennings et al. (2017) | | | | | x | Jennings et al. (2017) |
| *Pullenia osloensis* | Feyling-Hanssen, 1954 | x | Lloyd et al. (2011), Perner et al. (2011), Perner et al. (2013), Sheldon et al. (2016) | | | | | x | Sheldon et al. (2016) |
| *Reussoolina laevis* | (Montagu, 1803) | | | | | | | | |
| *Silicosigmoilina groenlandica* | Loeblich & Tappan, 1953 | | | | | | | | |
| *Stainforthia concava* | (Höglund, 1947) | | | x | Lloyd (2006), Jennings et al. (2017) | | | x | Jennings et al. (2017), Sheldon et al. (2016) |
| *Stainforthia feylingi* | Knudsen & Seidenkrantz, 1994 | | | x | Lloyd (2006), Lloyd et al. (2011), Perner et al. (2011), Perner et al. (2013), Jennings et al. (2017), Wangner et al. (2018) | | | x | Jennings et al. (2017), Sheldon et al. (2016) |
| *Trifarina fluens* | (Todd, 1948) | x | Lloyd (2006), Wangner et al. (2018) | | | x | Perner et al. (2013) | | |

## 13 Figures

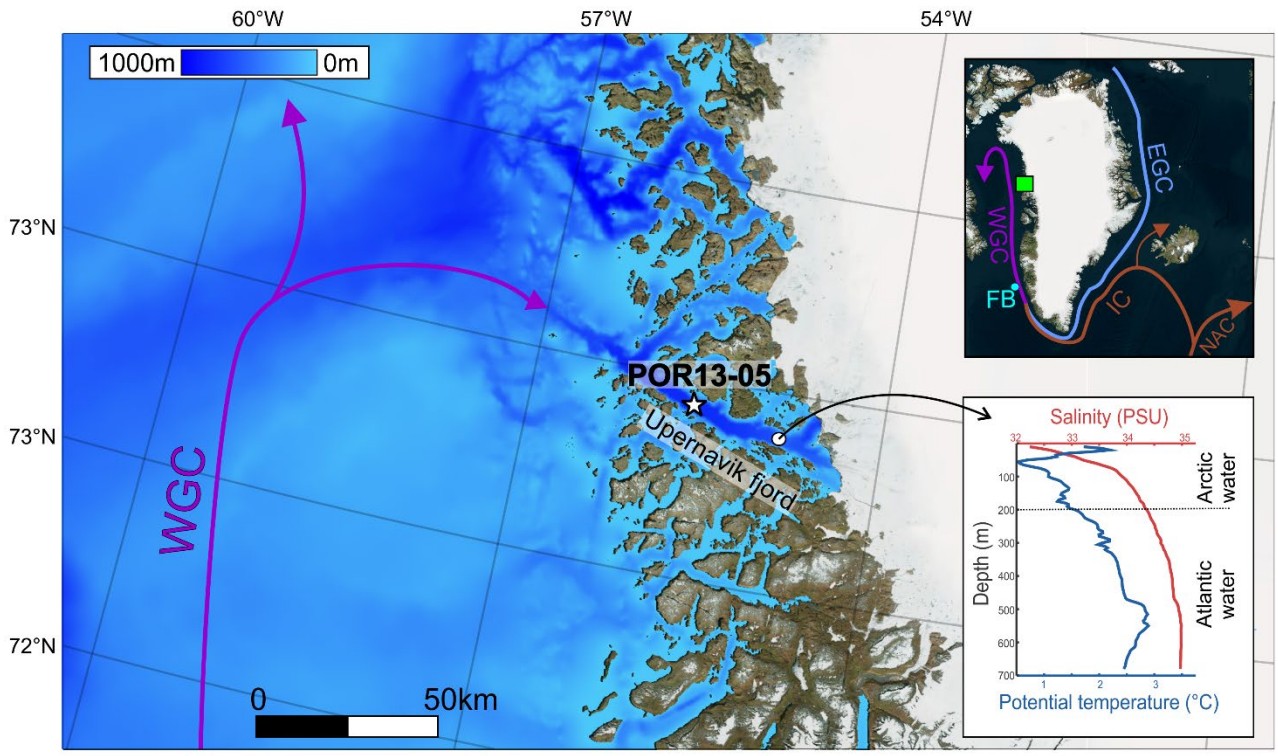

**Figure 1. Position of core site POR13-05 (star) in Upernavik Fjord together with the surrounding bathymetry (Morlighem et al., 2017). The inset figure indicates the ocean currents around Greenland (Bing satellite map, 2017). IC = Irminger Current, EGC= East Greenland Current, WGC = West Greenland Current, NAC= North Atlantic Current, FB= Fylla Banke. The green box in the inset marks the position of Upernavik Fjord, the light blue circle marks the position Fylla Banke. CTD profile is from Andresen et al. (2014).**

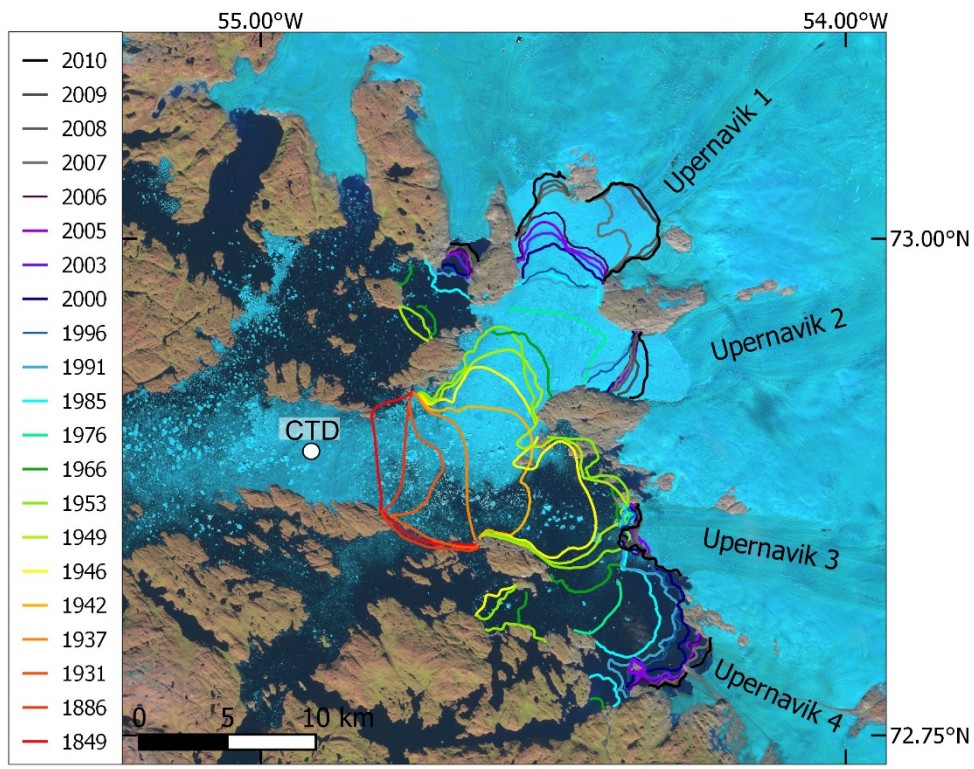

**Figure 2. Satellite image (2018-07-31, Sentinel-2B) with the historical record of front observations of Upernavik Isstrøm and location of CTD profile, presented in Fig. 1 (Andresen et al., 2014; Khan et al., 2013; Weidick, 1958).**

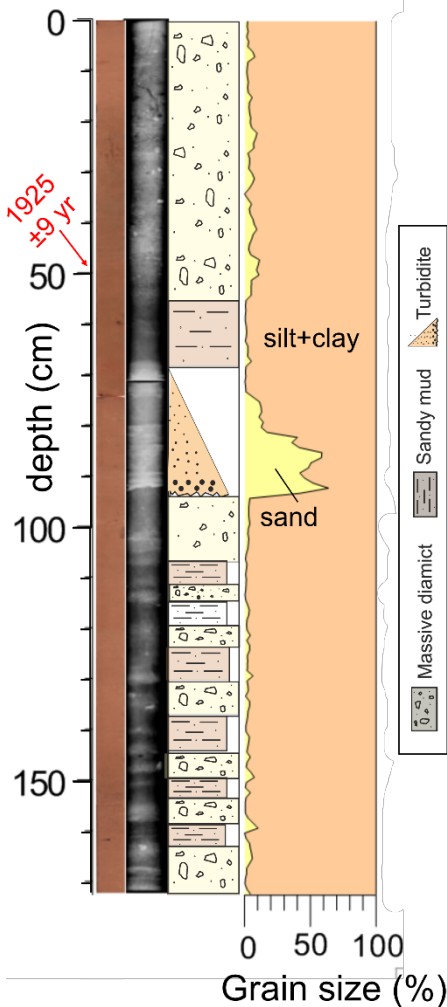

**Figure 3. Linescan and X-Ray image of sediment core POR13-05, together with grain-size measurements. The top 50 cm was dated with the [210]Pb method.**

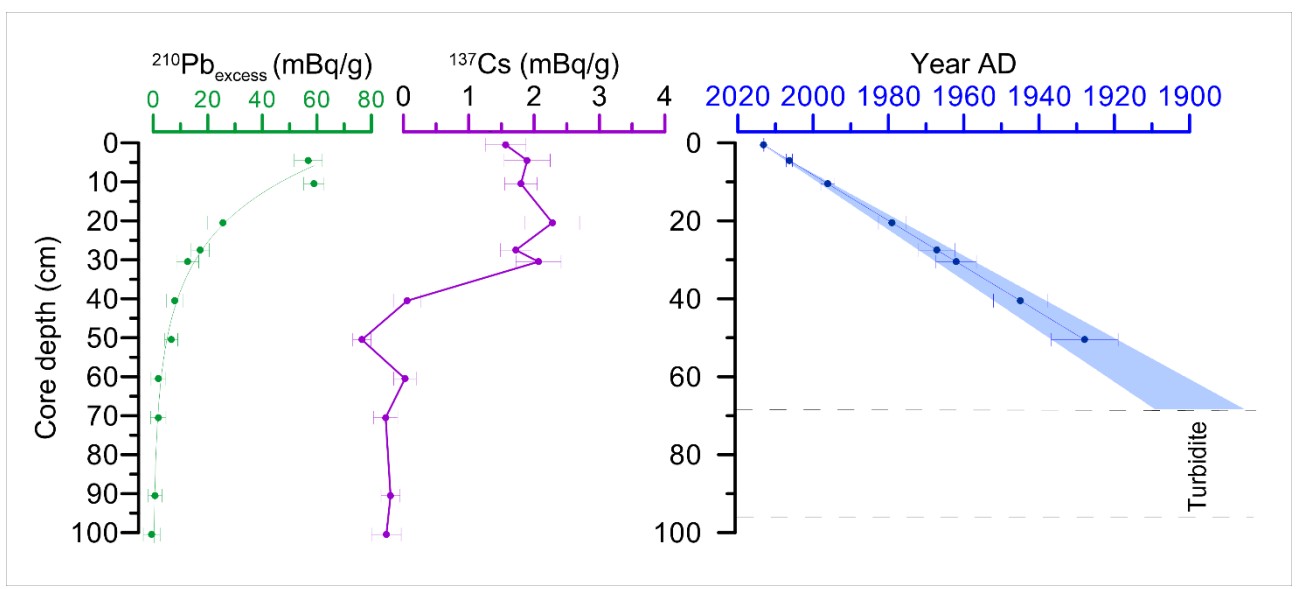

**Figure 4. Left) Measurements of $^{210}Pb_{xs}$ and $^{137}Cs$ in core POR13-05 according to core depth. Right) Age model calculated with the CS-CF model.**

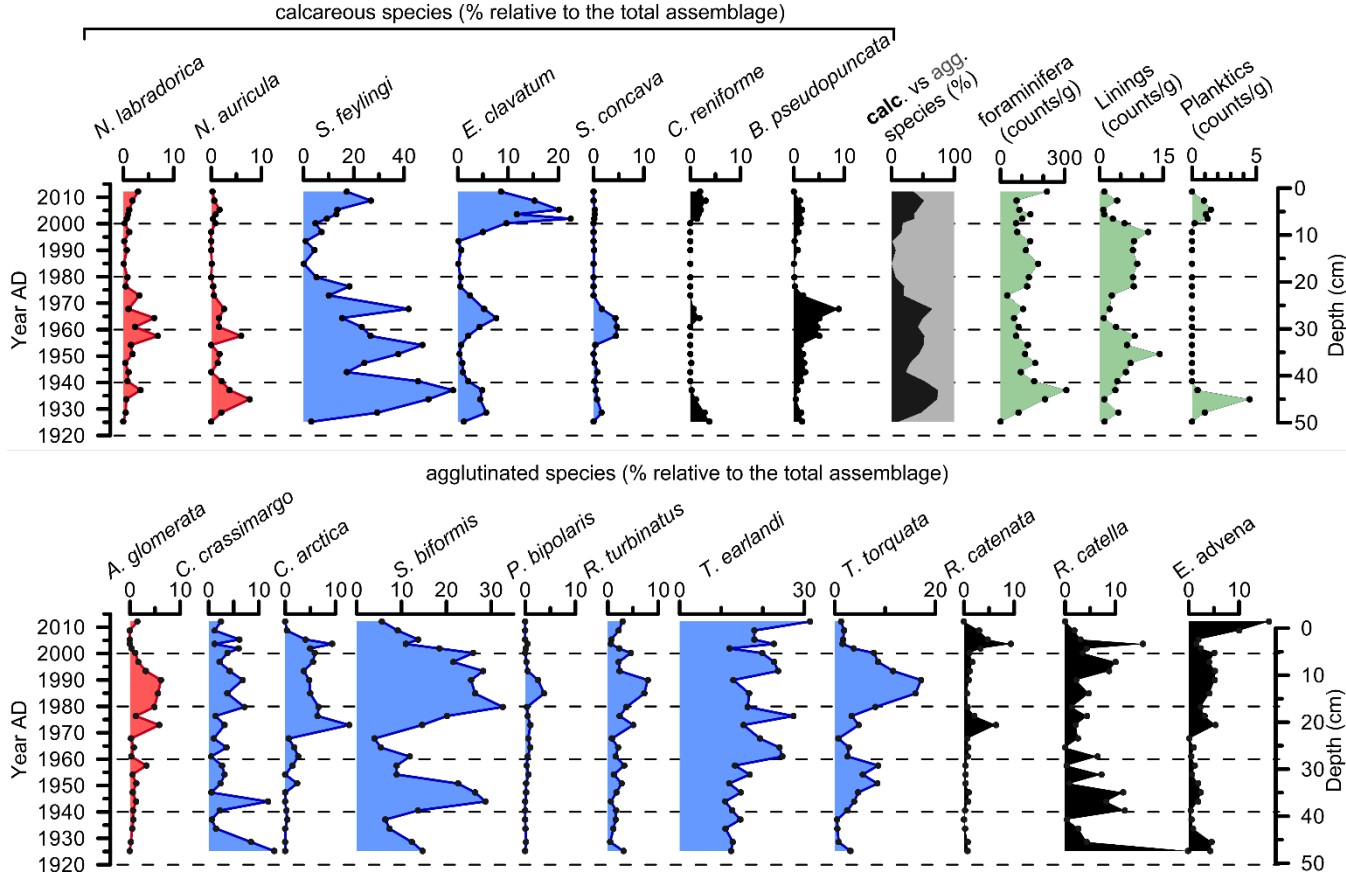

**Figure 5. Abundances of calcareous (top) and agglutinated benthic foraminiferal species (bottom), plotted versus age/depth in core POR13-05. Abundances are calculated relative to the total (calcareous and agglutinated) counts of benthic foraminifera. Red colors indicate (chilled) Atlantic water indicator species, blue colors represent water indicators. Only species representing >0.5% of the total assemblage are shown. Total count of foraminifera and organic linings per gram are also indicated (green, top). Stippled lines are a visual aid for comparing the different trends.**

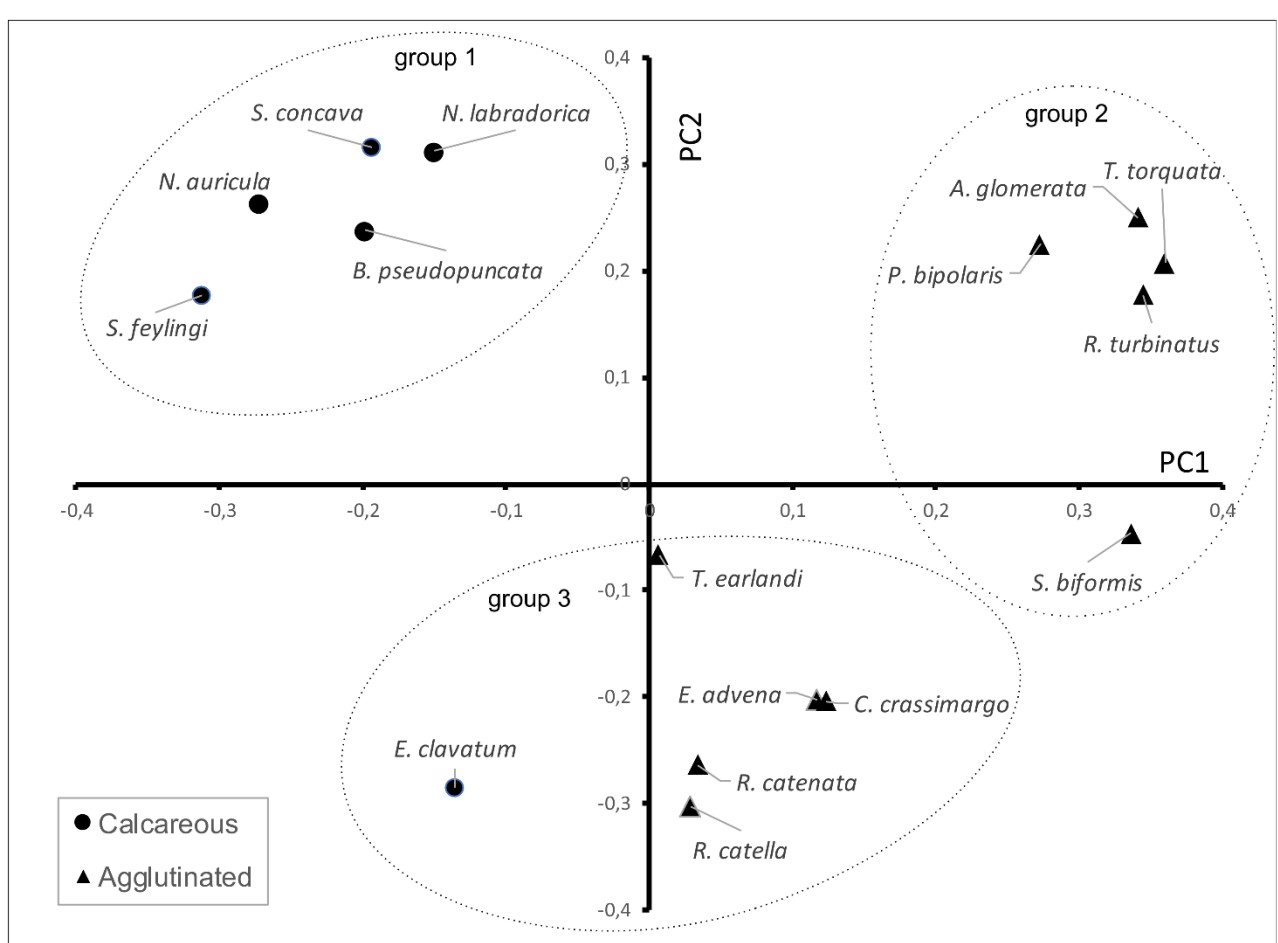

**Figure 6. Loadings plot resulting from the principal component analysis of the foraminiferal assemblages in POR13-05. PC1 and PC2 explain 28.6% and 16.8% of the variance, respectively. Only species representing >0.5% of the total amount of counted tests were taken into account for PCA analysis.**

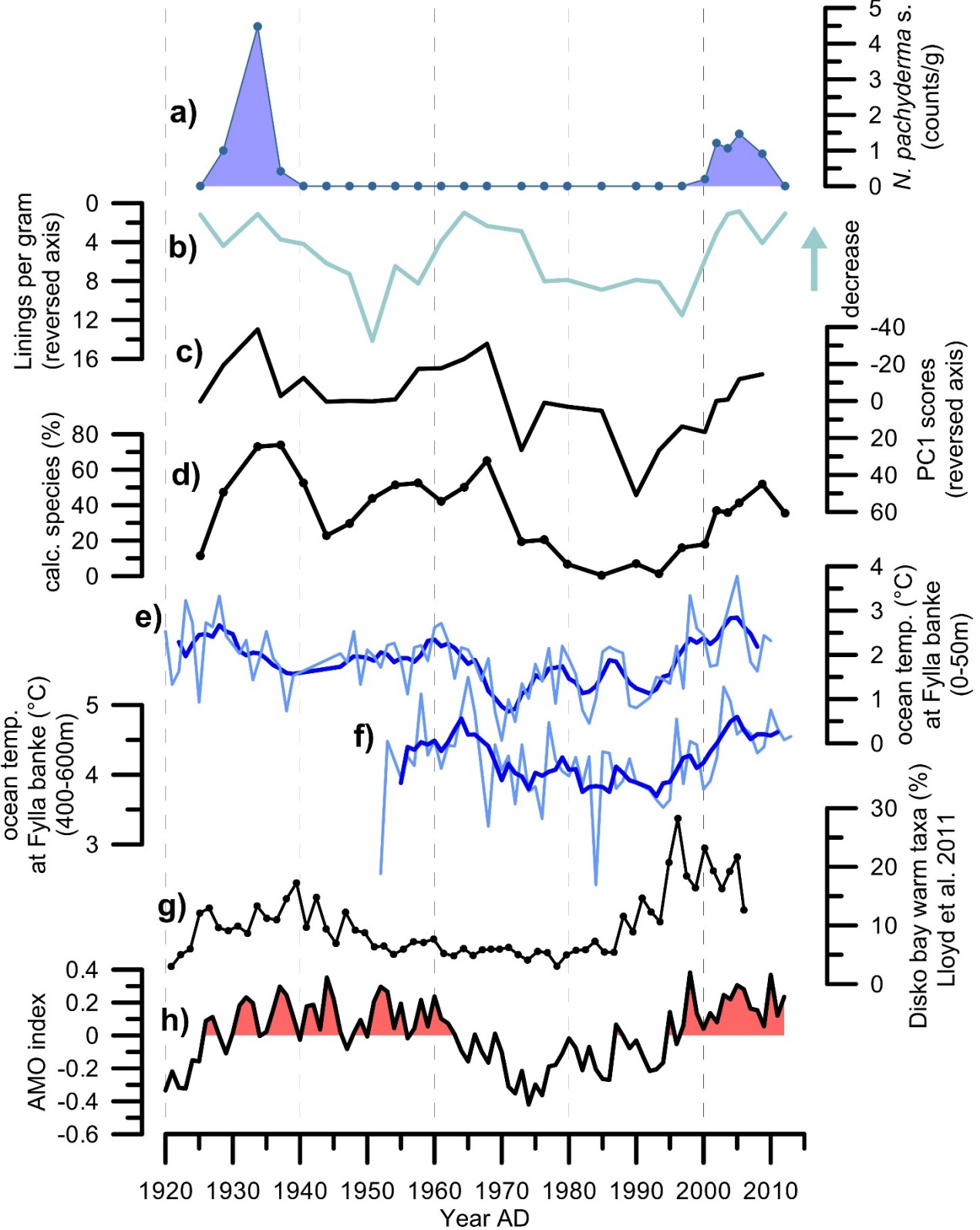

**Figure 7. Comparison of results from this study of core POR13-05 (a-d) with climate records (e, f) and reconstructions from West Greenland (g) and the Atlantic Multidecadal Oscillation (h). a) Abundance of *N. pachyderma* (sinistral) (counts/g). b) Number of organic linings per gram. c) Scores of PC1 based on analysis of species abundances. d) Percentage of calcareous foraminifera. e) Water temperatures (0-40m, June) measured from trawl surveys at Fylla Banke; data before 1950 were extended back to 1861 based observations from a wider area (Ribergaard et al., 2008). Dark blue line indicates the 3-year running average. f) Measured temperatures at Fylla Banke (400-600m) (Ribergaard et al., 2008). Dark blue line indicates the 3-year running average. g) Percentage of Atlantic water indicators from a benthic foraminifera study in Disko Bugt, West Greenland (Lloyd et al., 2011). h) AMO index is based on the definition by Enfield et al. (2001). Stippled lines are a visual aid for comparing the different records.**

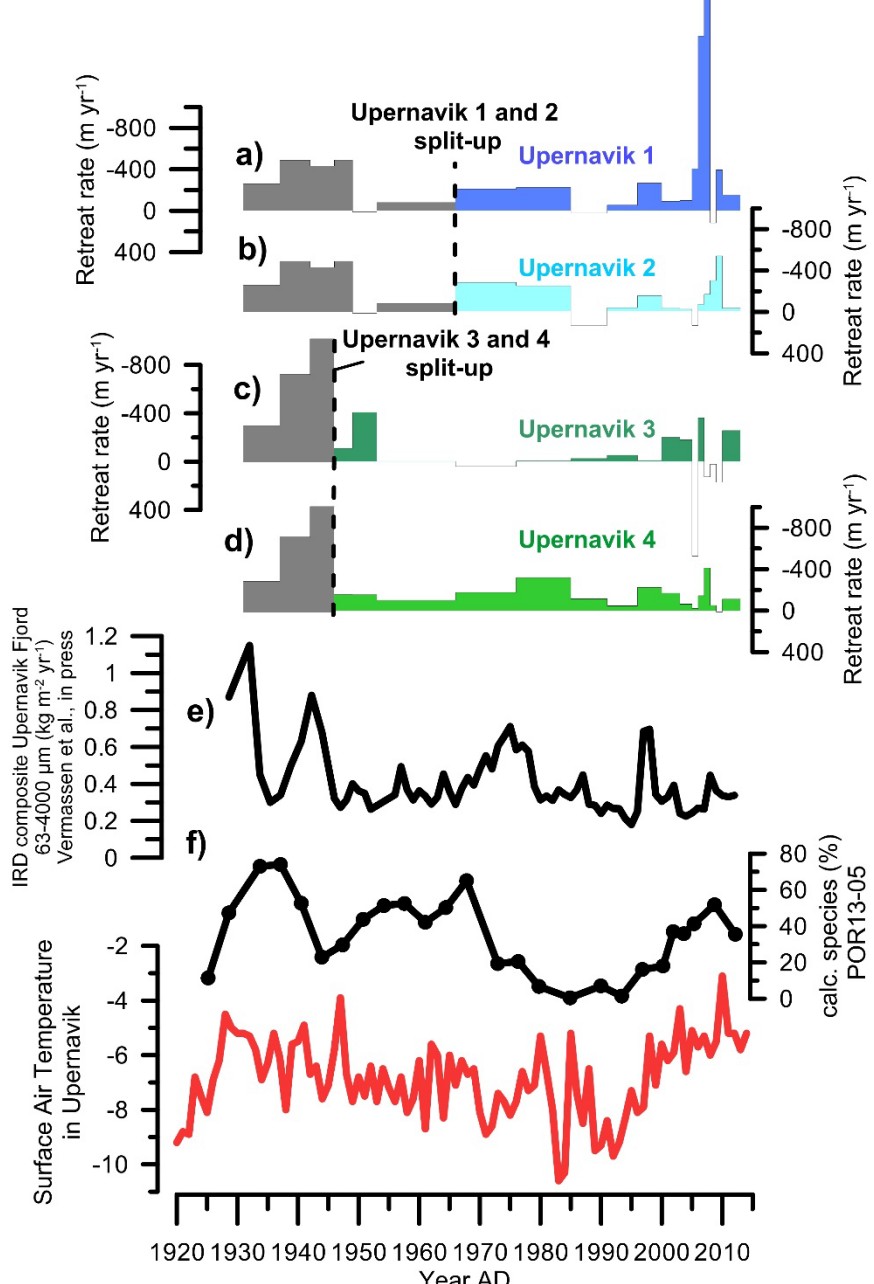

**Figure 8. Comparison of retreat rates of Upernavik Isstrøm's glaciers (Andresen et al., 2014) with reconstructed inflow of Atlantic water to Upernavik fjord (this study) and measured surface air temperatures in Upernavik (Cappelen, 2011). a) to d) Retreat rates of Upernavik 1, Upernavik 2, Upernavik 3, and Upernavik 4, respectively (Andresen et al., 2014). Grey colours indicate periods when the northern glaciers and southern glaciers were still joined together, stippled lines indicate when these split up. e) Composite record of IRD variability based on multiple sediment cores in Upernavik Fjord (Vermassen et al., in press). f) Percentage of calcareous foraminifera in core POR13-05, used as a proxy for Atlantic water inflow. g) Observed surface air temperatures in Upernavik (Cappelen, 2011).**