# Peer review of "A reconstruction of warm water inflow to Upernavik Isstrøm since AD 1925 and its relation to glacier retreat."

_Climate of the Past, 2018_

## Referee Comment (RC1) · Anonymous Referee #1 · 5 Feb 2019

Overall this is a very interesting paper that presents important new information regarding the interaction between ocean circulation and dynamics of a major northwest Greenland tidewater glacier. The paper presents new data from a sediment core reconstructing variations in ocean conditions, specifically the influx of relatively warm Atlantic Water into the fjord. The paper then discusses the possible link between this ocean forcing and variations in ice margin position over the last 100 years. The paper presents results of high scientific significance and quality. I have some questions/suggestions regarding some of the data presented as outlined below.

It is not immediately clear why the sedimentological information referred to (particle

[Figure]

size and IRD) has not been presented in this paper also. This data is often used to interpret glacier retreat rates – I suspect it would make for a much stronger final paper by combining with the faunal data. There may end up being overlap with paper submitted to Journal of Quaternary Science referred to in the manuscript.

A key aspect of this paper is the use for benthic foraminiferal data to interpret changes in bottom water conditions – specifically changes in relative importance of Atlantic Water flux across the shelf. The paper classifies the benthic foraminifera species into groups linked to bottom water conditions based on published studies (Atlantic influenced, Arctic and Indifferent). This is presented in a supplementary table. However, it is not always clear what the rationale for the choice of grouping is as for some species there is more than one potential group designation suggested in the literature. As this is such a key element of the paper I think it would be useful if the groupings and rationale were presented as part of the main paper. Or, alternatively, at least have a table in the main paper with species composition of the three groups stated, then refer to Supp Info for the more detailed rationale.

Presentation of the foraminiferal data. As above this is a fundamental part of the paper and the key interpretation and conclusions are based on the presentation of foraminiferal data. Calcareous foraminifera are presented separately from agglutinated species (Figure 5), however, the % calculation is based on the combined counts of calcareous and agglutinated specimens. As acknowledged in the paper the calcareous species are influenced by dissolution (most severely between 10 and 25 cm). Hence the relative proportions of the various species will be strongly influenced by preservation of the calcareous fauna rather than by bottom water preference. It would make sense to at least present an additional set of graphs showing the % abundance of agglutinated species based on agglutinated counts only. This would remove the influence of dissolution and would make trends in the proportion of Atlantic influenced vs Arctic influenced species clearer to identify. The same could be tried with the calcareous assemblage, though for parts of the core there would not be enough specimens to

present a robust % curve. If these plots do not show any clear/useful trends, then they could at least be included in the Supp Info section so that interested readers could see that the analysis had been done.

PCA analysis – linked to the point above, this seems to be largely driven by preservation. Group 1 is composed of entirely calcareous species (including a mix of Atlantic and Arctic water indicators) and Group 2 is entirely agglutinated species (also including a mix of Atlantic and Arctic indicators). This should be made clear in the paper. Based on this surely the PCA results simply show that foraminiferal preservation is a key control, certainly for PC1 at least. It would be interesting to try running PCA on the separated agglutinated fauna alone. This might actually provide more useful ecological information.

The key conclusion based on the faunal data is that bottom water conditions control dissolution – increased dissolution of calcareous fauna during times of colder Arctic Water influence, then reduced dissolution during periods of stronger Atlantic Water influx. This makes sense, but ought to also consider possible influence of sedimentation rate – increased sedimentation rate will lead to improved preservation of calcareous fauna (buried before dissolve). This has been identified from studies immediately in front of Jakobhavns Isbrae in Disko Bay (Lloyd et al 2005 and Lloyd 2006). You may conclude that sedimentation rate does not vary in your core, but I think this possibility ought to be presented.

I don't think these considerations will change the actual interpretation and conclusions of the paper, I think these are valid and generally supported by the data. Some of the points outlined above might help support your conclusions.

Section 5.2. Comparison with climatic records. This discussion is essentially based on the % Calcareous Fauna (ie preservation, Figure 7b). The manuscript suggests warm bottom waters during 1920-1960, but the basal sample from 1925 actually has a very low calcareous abundance (10%). This should be classified as cold water indicating...

The high calcareous percentages actually seem to stretch from 1930 – 1970 based on Fig 7b. So based on the data the warm bottom waters seem to be from 1930 – 1970. . . I think the description/discussion ought to reflect this more clearly. The early part of the 20th Century still seems to correlate reasonably well with previous reconstructions (Figure 8f) and the AMO (Figure 8g) – particularly given some error margins in age model generation.

Section 5.3 Retreat of Upernavik Isstrøm and ocean forcing. The record after 2000 seems mixed, seems to be major increase after 2005 for Upernavik 1 and 2 at least, while the ocean forcing is earlier. There is some discussion about the possible role of sea floor topography – it would be useful to provide a little more information here. Have previoius studies identified topographic variability as important in controlling retreat rates for the four different ice streams? Does fjord side wall configuration have an impact on retreat rates? It might also help to put a smoothing line through fig 8f so that it is more directly comparable with your ocean record resolution.

Minor points In the introduction the authors refer to previous research identifying rapid retreat and acceleration of tidewater glaciers in SE and NW Greenland – I think it would be worth making a distinction between NW and central west Greenland. The first studies identified this response from Jakobshavns Isbrae – this is probably better referred to as central west rather than northwest Greenland.

Reference by Ribergaard et al., 2008 does not seem complete.

Figure 8. It would be useful to say what the colours in panels a) to d) represent (times when the glaciers split up?).

Page 9 line 21: '. . .perturb glacier front', I wonder if it might be better to say '. . .reach glacier front'?

---

## Referee Comment (RC2) · Anonymous Referee #2 · 13 Feb 2019

Vermassen et al. present a nice paper on the palaeoceanography of a Greenlandic fjord. I my opinion, the paper is relevant and a good fit for Climate of the Past. However, there are some issues that need addressing and resolving before the paper can be published. I have outlined these below. I am happy to review a revised version of this manuscript.

 c I think the paper suffers from focussing solely on one proxy (benthic foraminifera), especially since relatively major oceanographic/environmental changes are inferred from the data. Supporting data, for example TIC/TOC, biogenic silica, stable isotopes of oxygen and carbon etc. would greatly strengthen the interpretations, especially those

regarding productivity. Although this may be outside the scope of this study, I still find it surprising that such relatively standard sedimentary analyses were omitted here.

• One paper – Vermassen et al. (submitted to JQS) is quoted repeatedly, with many of the authors overlapping with the present manuscript; however, this paper is under review and so presumably not available yet. Is the JQS paper focussing on the same core, and if so, what was the reason for publishing it separately rather than having it as one, stronger paper? I think this issue should be addressed, as relevant data from the JQS manuscript may strengthen the current one.

• Since your core was collected at 900 m depth, your benthics record bottom water conditions only. You mention planktonic foraminifera in passing, however [section 4.3; lines 11-12]. Although planktonics may be sparse, I still think this is important and should be expanded on, as they may give you a clue as to the validity of your overall interpretations.

• [section 5.1] The dissolution of calcareous foraminifera also depends on depth – for example, there are almost no calcareous foraminifera in deeper Baffin Bay waters.

• You identify organic linings in your samples and assign them to Elphidium excavatum. How confident are you that these linings are those of E. excavatum? Did you dissolve specimens of this species to check this? What about the linings of other planispiral species? If you get foraminiferal linings in your samples, you must have more than one type present – what are they? I think the link between linings and a specific foraminiferal species should be demonstrated more clearly, as this forms the basis of your argument regarding dissolution later on [section 5.1], and, especially, the link to Atlantic water inflow.

• Also regarding linigs, the abundance in Fig. 5 are rather on the low side (max. 15 linings/g] – is this correct? Does this include all linings or just those of planispiral species? How do you make the leap between dissolution and lining presence, especially in periods that have plenty of calcareous proportions but also the highest rates of

linings (e.g., 1920-1960 on Fig. 5)?

• Inflow of Atlantic waters into Baffin Bay and adjacent regions are inferred in previous palaeo-studies (e.g., Knudsen et al. 2012, Boreas). How similar or dissimilar are the benthic assemblages in these studies compared to the present one?

• The inference of nutrient levels in the fjord [section 5.1, line 24 onwards] is tentative in my opinion, since you don't have other palaeoproductivity indicators (TOC, d13C) to support this notion. True, there are some species in your record which indicate high flux of organic matter to the seafloor, but this is mostly N. labradorica. I think these inferences between dissolution, Atlantic water inflow, and nutrients should be done with greater caution, therefore.

• Fig. 5: this figure is a bit confusing and should be modified for ease of reading. For example, are the abundances shown relative or absolute? What are the stippled lines (very faint!)? I suggest adding lines or points to the silhouette graphs so the number of samples/datapoints can be seen more clearly.

• Unless I missed it, a list of all species found, including taxonomic designations, should be added to the paper.

---

## Editor Comment (EC1) · Alberto Reyes (Editor) · 4 Mar 2019

Dear Dr. Vermassen and co-authors,

The discussion period for your manuscript is now over, and two reviewers have posted comments. Both reviewers were positive about the manuscript and the scientific value of the underlying research question. The reviewers made several constructive suggestions for presentation and critical consideration of the data, as well as some points to consider in interpreting your data. There is also some concern expressed about overlap with your manuscript currently under consideration with JQS, which could potentially be addressed with some additions to the Fig 3 plots. My sense is that you can

address the concerns of the reviewers with relatively minor revision.

Please respond to the reviewer comments in the online Discussion forum. If you make any changes to your manuscript, please clearly indicate the nature of these changes in your response. Once I have reviewed your response, I fully anticipate inviting you to submit a revised manuscript for a final decision.

Sincerely, Alberto Reyes

––––––––––––––––––––––––––

---

## Author Comment (AC1) · 27 Mar 2019

We thank the reviewer for their valuable comments and suggestions, and find them very helpful for improving and clarifying the manuscript.

*It is not immediately clear why the sedimentological information referred to (particle size and IRD) has not been presented in this paper also. This data is often used to interpret glacier retreat rates – I suspect it would make for a much stronger final paper by combining with the faunal data. There may end up being overlap with paper submitted to Journal of Quaternary Science referred to in the manuscript.*

Sedimentological data (i.e. IRD) is indeed often used in combination with foraminiferal data to assess the relationship between oceanic changes and glacier activity (iceberg calving). A standard approach would be to show the grain-size/IRD data from this core, present it as a proxy for glacier calving and then attempt to interpret their relationship. Thus I can see it comes across as somewhat strange that I did not present the IRD record in this paper.

The main reason for this is that when we investigated IRD patterns in the fjord (based on multiple sediment cores) we found that the relationship between IRD and glacier behavior is not as straightforward as is often assumed. Therefore we decided to write an article specifically discussing the relation between IRD and glacier behavior (i.e. the JQS paper), and as such IRD was not addressed in the current article. We think that in this this way the article presented here can focus more clearly on the question 'does/did ocean variability affect glacier behavior. Also, in this study we rely on historical glacier front positions available, which are probably more accurate than interpretation of the often complex relationship between glacier dynamics and IRD (see section above). Nevertheless, we agree that it is still be worthwhile to show the reader the IRD data, so I have included this in the revised article (figure 7).

*Presentation of the foraminiferal data. As above this is a fundamental part of the paper and the key interpretation and conclusions are based on the presentation of foraminiferal data. Calcareous foraminifera are presented separately from agglutinated species (Figure 5), however, the % calculation is based on the combined counts of calcareous and agglutinated specimens. As acknowledged in the paper the calcareous species are influenced by dissolution (most severely between 10 and 25 cm). Hence the relative proportions of the various species will be strongly influenced by preservation of the calcareous fauna rather than by bottom water preference. It would make sense to at least present an additional set of graphs showing the % abundance of agglutinated species based on agglutinated counts only. This would remove the influence of dissolution and would make trends in the proportion of Atlantic influenced vs Arctic influenced species clearer to identify. The same could be tried with the calcareous assemblage, though for parts of the core there would not be enough specimens to present a robust % curve. If these plots do not show any clear/useful trends, then they could at least be included in the Supp Info section so that interested readers could see that the analysis had been done.*

*A key aspect of this paper is the use for benthic foraminiferal data to interpret changes in bottom water conditions – specifically changes in relative importance of Atlantic Water flux across the shelf. The paper classifies the benthic foraminifera species into groups linked to bottom water conditions based on published studies (Atlantic influenced, Arctic and Indifferent). This is presented in a supplementary table. However, it is not always clear what the rationale for the choice of grouping is as for some species there is more than one potential group designation suggested in the literature. As this is such a key element of the paper I think it would be useful if the groupings and rationale were presented as part of the main paper. Or, alternatively, at least have a table in the main paper with species composition of the three groups stated, then refer to Supp Info for the more detailed rationale.*

We agree that the grouping rationale was indeed too briefly explained. The table that lists all identified species and references to which ecological habit that species have been previously assigned is now moved to the main document. For species that have been assigned as both Atlantic and Arctic water indicators, we used the most recent studies (but also listing the contradicting references). It is important to note that we use these previously attributed ecological preferences mostly as a tool to investigate our dataset (i.e. to investigate whether species with similar proposed environmental preferences indeed display similar 'groupings' within Upernavik fjord. We then found that the general variation within the dataset is largely driven by the proportion of calcareous vs. agglutinated taxa present (as seen by results of the PCA analysis). PCA analysis run for each group (calcareous taxa, agglutinated taxa) seems inconclusive with regards to grouping of species by environmental preferences (Atlantic/Arctic) and we have now shown this analysis in the supplementary information (i.e. the requested abundances and PCA plots). Within the calcareous assemblage, a lack of grouping according to environmental preference can probably be explained because the info provided by species shifts is limited since calcareous species are nearly absent in a large interval. Also, little regard has been given in the literature to the sensitivity of agglutinated assemblages to environmental conditions (one of the few studies is Lloyd et al., 2006), and only one species in our agglutinated assemblage (>0.5% of total abundance) has been proposed as an indicator of Atlantic waters. Therefore, in our discussion the interpretation relies predominantly on the calcareous/agglutinated ratio, rather than the environmental preferences assigned to the species as proposed by previous work. I have rewritten parts of the results and discussion, which I hope will make this reasoning more clear.

*I don't think these considerations will change the actual interpretation and conclusions of the paper, I think these are valid and generally supported by the data. Some of the points outlined above might help support your conclusions.*

*The key conclusion based on the faunal data is that bottom water conditions control dissolution – increased dissolution of calcareous fauna during times of colder Arctic Water influence, then reduced dissolution during periods of stronger Atlantic Water influx. This makes sense, but ought to also consider possible influence of sedimentation rate – increased sedimentation rate will lead to improved preservation of calcareous fauna (buried before dissolve). This has been identified from studies immediately in front of Jakobhavns Isbrae in Disko Bay (Lloyd et al 2005 and Lloyd 2006). You may conclude that sedimentation rate does not vary in your core, but I think this possibility ought to be presented.*

We agree and have included a line about the effect variations in sedimentation rate may have on the preservation of calcareous fauna.

*Section 5.2. Comparison with climatic records. This discussion is essentially based on the % Calcareous Fauna (ie preservation, Figure 7b). The manuscript suggests warm bottom waters during 1920-1960, but the basal sample from 1925 actually has a very low calcareous abundance (10%). This should be classified as cold water indicating...*

This has been changed accordingly in the manuscript.

*The high calcareous percentages actually seem to stretch from 1930 – 1970 based on Fig 7b. So based on the data the warm bottom waters seem to be from 1930 – 1970... I think the description/discussion ought to reflect this more clearly. The early part of the 20th Century still seems to correlate reasonably well with previous reconstructions (Figure 8f) and the AMO (Figure 8g) – particularly given some error margins in age model generation.*

We agree and have rewritten this section of the discussion accordingly.

*Section 5.3 Retreat of Upernavik Isstrøm and ocean forcing. The record after 2000 seems mixed, seems to be major increase after 2005 for Upernavik 1 and 2 at least, while the ocean forcing is earlier. There is some discussion about the possible role of seafloortopography–it would be useful to provide a little more information here. Have previous studies identified topographic variability as important in controlling retreat rates for the four different ice streams? Does fjord side wall configuration have an impact on retreat rates? It might also help to put a smoothing line through fig 8f so that it is more directly comparable with your ocean record resolution.*

Subglacial topography indeed plays a major role in controlling the dynamics of marine terminating glaciers and there is some research available for Upernavik Isstrøm. Sidewall configuration plays a role but is less well understood. I have expanded this section in the discussion, but to address all of the literature on the subject would beyond the scope of this study. We hope the reviewers agree that we have found the right balance in the revised manuscript.

*Minor points In the introduction the authors refer to previous research identifying rapid retreat and acceleration of tidewater glaciers in SE and NW Greenland – I think it would be worth making a distinction between NW and central west Greenland. The first studies identified this response from Jakobshavns Isbrae – this is probably better referred to as central west rather than northwest Greenland..*

We agree and included this distinction in the introduction.

*Reference by Ribergaard et al., 2008 does not seem complete*

Reference is completed.

Figure 8. It would be useful to say what the colours in panels a) to d) represent (times when the glaciers split up?).

These colours indeed refer to the periods when glaciers were still together (grey) and when they split. We describe this better in the caption now.

*Page 9 line 21: '...perturb glacier front', I wonder if it might be better to say '...reach glacier front'?*

Agreed and this has been rewritten.

---

## Author Comment (AC2) · 27 Mar 2019

We thank the reviewer for their comments and suggestions. We believe that incorporating these comments will lead to an improved manuscript.

*Vermassen et al. present a nice paper on the palaeoceanography of a Greenlandic fjord. I my opinion, the paper is relevant and a good fit for Climate of the Past. However, there are some issues that need addressing and resolving before the paper can be published.*

*1. I think the paper suffers from focusing solely on one proxy (benthic foraminifera), especially since relatively major oceanographic/environmental changes are inferred from th edata. Supporting data, for example TIC/TOC,biogenic silica, stable isotopes of oxygen and carbon etc. would greatly strengthen the interpretations, especially those regarding productivity. Although this may be outside the scope of this study, I still find it surprising that such relatively standard sedimentary analyses were omitted here.*

We certainly agree that multi-proxy analysis can add value to sediment core analysis. Unfortunately, due to time constraints as well as limited available sediment, we cannot complete further additional analysis.

*2. One paper – Vermassen et al. (submitted to JQS) is quoted repeatedly, with many of the authors overlapping with the present manuscript; however, this paper is under review and so presumably not available yet. Is the JQS paper focussing on the same core, and if so, what was the reason for publishing it separately rather than having it as one, stronger paper? I think this issue should be addressed, as relevant data from the JQS manuscript may strengthen the current one.*

The quoted article (JQS) has meanwhile been accepted for publication and is in production. The JQS article investigates IRD variations from multiple cores in Upernavik fjord, also including this core. The main reason that the IRD data is not incorporated in this article (Climate of the Past) is that we found that the relationship between IRD and glacier behavior is not as straightforward as is often assumed. Therefore we decided to write a 'proxy-evaluation' article that specifically discusses the relation between IRD and glacier behavior (i.e. the JQS paper), and this is why the IRD is not addressed in this paper. This allows this Climate of the Past paper to focus more clearly on the question 'does/did ocean variability affect glacier behavior'. Nevertheless, I agree that it would still be good to at least also show the IRD data this article, so I include this now in Fig. 7.

*3. Since your core was collected at 900 m depth, your benthics record bottom water conditions only. You mention planktonic foraminifera in passing, however [section 4.3; lines 11-12]. Although planktonics may be sparse, I still think this is important and should be expanded on, as they may give you a clue as to the validity of your overall interpretations.*

We agree that the way the planktonic species were mentioned was not very clear. Some planktonics were found, but in very low abundances. I expand this section and add the variability in total no. of planktonic foraminifera per gram to Figs. 5 and 7.

*4. The dissolution of calcareous foraminifera also depends on depth – for example, there are almost no calcareous foraminifera in deeper Baffin Bay waters.*

We have added a line about this to the manuscript.

*5. You identify organic linings in your samples and assign them to Elphidium excavatum. How confident are you that these linings are those of E. excavatum? Did you dissolve specimens of this species to check this? What about the linings of other planispiral species? If you get foraminiferal linings in your samples, you must have more than one type present – what are they? I think the link between linings and a specific*

*foraminiferal species should be demonstrated more clearly, as this forms the basis of your argument regarding dissolution later on [section 5.1], and, especially, the link to Atlantic water inflow.*

We agree that identifying a species solely based on organic lining is too tentative and removed mentioning E. clavatum as the main species. We did not intend to present the variability in organic linings as the main argument for deriving an effect of dissolution on the faunal assemblage. Rather, we derive the importance of dissolution based on the the distinct variation between agglutinated/calcareous species and the fact that this has been reported as a very important processes in Arctic waters/fjords. I have rewritten the discussion to make it more clear that the variation in organic linings is not the main argument for the effect of dissolution, but is rather one element that supports our interpretation.

6. *Also regarding linings, the abundance in Fig. 5 are rather on the low side (max. ´ 15 linings/g] – is this correct? Does this include all linings or just those of planispiral species? How do you make the leap between dissolution and lining presence, especially in periods that have plenty of calcareous proportions but also the highest rates of linings (e.g., 1920-1960 on Fig. 5)?*

The numbers of linings per gram are correct. Tests that were partly dissolved but still showed enough features to be identified were identified, otherwise they were counted as organic linings (the amount of partly dissolved test was very low tough, most linings showed no remnants of carbonate test). The organic linings were nearly all planispiral, but some unidentified organic tubes were also included in the count number (very few however). For thorough analysis of organic linings the samples should be prepared as playnological slides, but here we use them simply as a (rough) indication of dissolution of the calcareous assemblage.

7. *Inflow of Atlantic waters into Baffin Bay and adjacent regions are inferred in previous palaeo-studies (e.g., Knudsen et al. 2012, Boreas). How similar or dissimilar are the benthic assemblages in these studies compared to the present one.*

The benthic assemblage is very similar to that of Lloyd et al., (2012) and this has now been cited in the text This can be expected since this study is most comparable study available with regard to proximity, investigated time interval, water depth and sample preparation.

8. *The inference of nutrient levels in the fjord [section5.1,line 24onwards] is tentative in my opinion, since you don't have other palaeoproductivity indicators (TOC, d13C) to support this notion. True, there are some species in your record which indicate high flux of organic matter to the seafloor, but this is mostly N. labradorica. I think these inferences between dissolution, Atlantic water inflow, and nutrients should be done with greater caution, therefore.*

I agree this interpretation would have indeed be stronger with additional paleo-productivity proxies and the link with productivity is somewhat speculative. I have rewritten this section to incorporate a bit more caution with regard to the interpretation. In general we still believe that this is a point worth suggesting, in particular since these data might confirm recent research that has showed a strong link between Atlantic water inflow and (modern) productivity (Meire et al., 2017) .

9. *This figure is a bit confusing and should be modified for ease of reading. For ´ example, are the abundances shown relative or absolute? What are the stippled lines (very faint!)? I suggest adding lines or points to the silhouette graphs so the number of samples/datapoints can be seen more clearly.*

We adjust the figure so it is clear these are absolute abundances. The stippled lines are merely for visual support to compare the different abundances. We add a symbols for data points.

*10. Unless I missed it, a list of all species found, including taxonomic designations, should be added to the paper.*

This list is now present and placed main this the main document (Table 1).

---

## Author Comment (AC3) · 2 Apr 2019

Dear Editor

We have adressed the reviewer's comments and suggestions. We look forward to submitting the revised manuscript upon your invitation.

Sincerely Flor Vermassen